# FORMCRAFT: BEYOND DOCUMENTS— BENCHMARKING FORM INTELLIGENCE

## ABSTRACT

Current document AI benchmarks, focused on isolated OCR and span-based Visual Question-Answering (VQA) tasks, have reached a plateau. They fall short in evaluating the critical abilities of structural reconstruction and relational reasoning, which are essential for understanding real-world forms. We introduce FORMCRAFT, a structure-and relation-centric benchmark for form intelligence in vision–language models (VLMs). Unlike OCR or DocVQA-centric evaluations, FORMCRAFT operationalizes a three-level taxonomy: Content Modality (L1), Layout Structure (L2), and Semantic Relation (L3), into targeted tasks and structure-aware metrics. On real-world forms with professional annotation, we find that recognition is relatively strong while hierarchical reconstruction and *cross-field consistency* remain challenging across popular open-source and proprietary models. Upon paper acceptance, we will release the dataset and annotation schema to standardize future research.

## 1 INTRODUCTION

The digital transformation of business processes has placed automated form understanding at the forefront of document AI research. Forms serve as the primary interface for data collection and processing across diverse domains, from healthcare records (Mistry & Arzeno, 2023) to financial services and government documents (Häyrinen et al., 2008; 15489-1, 2016). While traditional forms consisted primarily of text fields and simple layouts, modern forms have evolved into complex, multi-modal documents incorporating interactive elements, dynamic layouts, and rich visual components (Xu et al., 2021). This evolution demands a paradigm shift in automated form understanding—moving beyond simple text extraction to interpreting the functional hierarchy and implicit logic encoded in the layout.

Despite significant advances in computer vision and natural language processing, current form understanding benchmarks predominantly emphasize Optical Character Recognition (OCR) (Park et al., 2019; Wen et al., 2018; Fu et al., 2024; Liu et al., 2024b) accuracy or general Visual Question Answering (VQA) tasks (Wang et al., 2023c; Fu et al., 2024; Liu et al., 2024b). These existing evaluations address important yet isolated capabilities, providing limited insight into the nuanced challenges presented by modern forms. Contemporary forms demand more sophisticated approaches that extend beyond text extraction, such as understanding spatial relationships between elements, interpreting cross-modal dependencies, and reasoning about complex hierarchical structures. Existing evaluation frameworks may not fully capture the holistic understanding required for real-world form processing applications, where integrated comprehension across multiple dimensions is essential.

Furthermore, while current Visual-Language Models demonstrate promising results on conventional OCR tasks and simple VQA scenarios, their ability to process and understand forms as coherent, structured documents remains less thoroughly evaluated. This represents an intriguing opportunity, particularly as organizations increasingly rely on automated systems for processing complex forms. Such systems ideally would demonstrate human-like understanding of form structure, content relationships, and semantic meaning. A holistic evaluation framework that addresses these integrated aspects of form understanding would help bridge the gap between isolated capability testing and the requirements of practical document processing applications.

To address these challenges, we propose FORMCRAFT, a comprehensive benchmarking framework with a three-level taxonomy and corresponding evaluation methods for form understanding capabilities. FORMCRAFT brings the following contributions:

- Propose a systematic three-level taxonomy: (L1) Content Modality, (L2) Layout Structure, and (L3) Semantic Relation. This allows for a granular, bottom-up evaluation of a model's capabilities, from basic recognition to complex reasoning. This taxonomy provides a systematic foundation for analysis that disentangles the complex, interdependent challenges of form intelligence, enabling more precise evaluation and targeted improvements in model capabilities.

- Develop comprehensive benchmark that operationalizes the proposed taxonomy through carefully designed evaluation tasks reflecting real-world form processing challenges.

- Conduct extensive experiments with state-of-the-art VLMs, revealing critical limitations in handling complex form structures and performing higher-order reasoning tasks. The results quantify the significant gap between current AI capabilities and real-world requirements, providing concrete directions for future research in document understanding.

- Demonstrate the framework's versatility by showing how it can be applied to existing form datasets, offering a standardized evaluation methodology for the document AI community.

- Open release: A professionally annotated, multi-modal form dataset with **field / group / nested-group** hierarchy, modality subtypes, and labeled relations will be released and open-sourced upon paper acceptance.

## 2 RELATED WORK

Prior work largely evaluates token-level reading (OCR) or span-level question answering (DocVQA). Forms add functional hierarchy and cross-field constraints; thus structure- and relation-centric evaluation is required rather than text-centric proxies.

### 2.1 BENCHMARKS FOR OCR

OCR remains a foundational component of document AI, with early benchmarks and datasets primarily designed to evaluate text recognition accuracy in isolation. Traditional datasets (Huang et al., 2019; Cheng et al., 2022; Veit et al., 2016) mostly focus on handwritten, receipt-based, or scene text recognition, offering annotated samples for tasks like character detection and transcription. Similarly, synthetic datasets (Gupta et al., 2016; Kim et al., 2022) provide large-scale, controlled environments for training OCR systems across languages and domains. Other efforts, such as CTW (Yuan et al., 2019) and TextOCR (Singh et al., 2021), extend this to real-world scenarios with diverse text appearances, while multilingual datasets like Chinese-OCR (Chen et al., 2021) addresses language-specific challenges. These datasets have been instrumental in advancing OCR robustness, yet their scope remains narrow, focusing predominantly on text extraction rather than the structural or semantic complexities of forms.

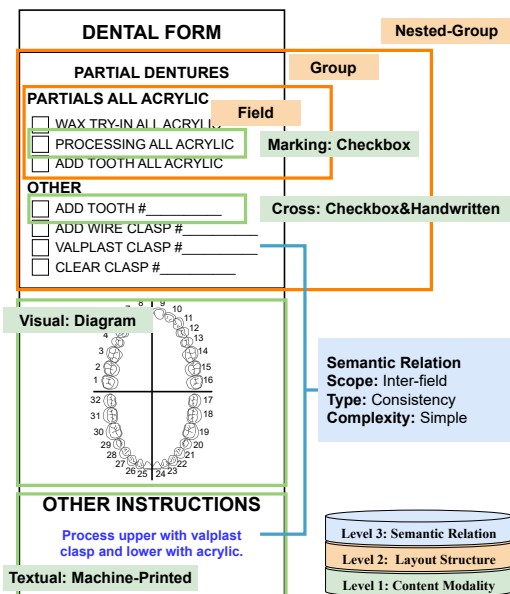

Figure 1: **FORMCRAFT overview.** Three levels: L1 content modality (green), L2 layout structure (orange), L3 semantic relation (blue). Annotations include field/group/nested-group trees, modality subtypes, and relation triples with scope/type/complexity.

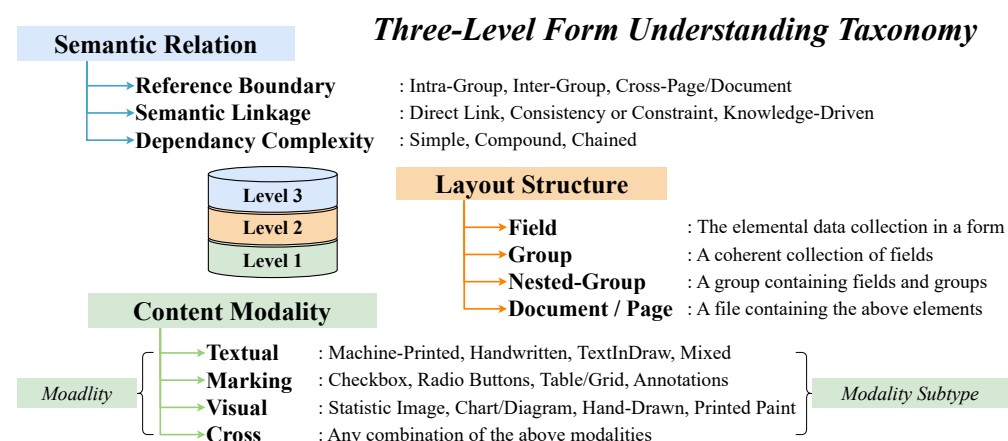

Figure 2: **Overview of FORMCRAFT taxonomy.** Our framework comprises three hierarchical levels: (a) Content Modality (green), which evaluates recognition capabilities across different information representation types; (b) Layout Structure (orange), which assesses understanding of form structures and spatial relationships; and (c) Semantic Relationship (blue), which measures cross-reference comprehension and logical dependencies between form elements. This three-level taxonomy enables systematic assessment of form intelligence across increasing levels of complexity.

## 2.2 VISUAL QUESTION ANSWERING

VQA plays a critical role in evaluating visual-language intelligence. Conventional VQA tasks focus on real-world images including general scene understanding (Marino et al., 2019; Schwenk et al., 2022; Zhu et al., 2016; Acharya et al., 2019; Shao et al., 2019; Goyal et al., 2017), object grounding (Shao et al., 2019; Gupta et al., 2022; Yu et al., 2016; Wang et al., 2023b; 2024; 2023a; Gao et al., 2023c), and captioning (Dai et al., 2024; Shao et al., 2019; Sidorov et al., 2020; Zhang et al., 2023; Liu et al., 2024a; Chen et al., 2023). Several benchmarks test AI's understanding of structured content like charts (Methani et al., 2020; Kahou et al., 2017; Li et al., 2024; Masry et al., 2023) and documents (Mathew et al., 2021; Laurençon et al., 2024), and yet these chart-based tasks typically address only single tables or diagrams, primarily for information extraction, while document-focused benchmarks often lack additional human annotation. With recent advances in large vision-language models, new benchmarks (Hudson & Manning, 2019; Gao et al., 2023a;b; Chen et al., 2024; Masry et al., 2022; Kazemi et al., 2024) increasingly test complex reasoning abilities. However, existing form understanding benchmarks (Jaume et al., 2019; Liu et al., 2024b; Fu et al., 2024) still focus broadly on text-centric tasks rather than addressing the specific structural and semantic challenges of forms, such as hierarchical layouts or cross-modal dependencies that require multi-hop reasoning capabilities.

## 3 FORMCRAFT FRAMEWORK

In this work, we propose a taxonomy for classifying form properties to enable more accurate and targeted evaluation across different intelligence levels. FORMCRAFT taxonomy consists of three fundamental levels: (a) Content modality, (b) Layout structure and (c) Semantic relation.

### 3.1 WHAT IS A FORM?

We define a "form" to be a structured document, intentionally designed as an interactive medium to elicit, collect, and organize specific information from a user. Unlike general documents that primarily disseminate information (e.g., articles, reports), a form's primary purpose is data acquisition. This purpose yields distinctive and challenging characteristics:

- Intentional layout that encodes a visual hierarchy of fields, groups, and sections, guiding user action.

- Inherent multimodality, mixing machine-printed instructions, interactive elements (checkboxes/radio), dedicated spaces for handwritten entries, and diagrams/tables.
- Implicit logic, where layout-driven constraints and dependencies (e.g., activation, validation, cross-field consistency, signatures) govern how inputs are provided and interpreted.

**Why forms are *not* generic documents**   Generic document understanding (e.g., plain articles) typically centers on local token recognition and coarse layout cues. In contrast, forms impose **structure- and relation-centric** requirements that standard DocVQA-style settings do not capture. Four key mismatches are summarized below:

In generic documents, spatial proximity and reading order often suffice to recover topical structure. Forms, however, encode *functional* hierarchies (field → group → nested-group) that need to be *reconstructed*, not merely detected. Two fields can be adjacent yet belong to different functional parents.

Unlike generic documents dominated by machine-printed text, forms mix handwritten notes, markings (checkbox/radio/ink), icons, and fine-grained visual affordances. Information is often not in the text string at all (e.g., a ticked box overrides a default). This demands modality-aware extraction and conflict resolution between modalities.

Many form answers are relational: an entry is valid only if a prerequisite box is checked, or a handwritten correction supersedes pre-filled text. Such cross-field dependencies are non-local and frequently cross-page, exceeding typical span-based QA.

Therefore, progress on forms requires recovering hierarchical structure and enforcing cross-field constraints under multi-modal evidence. Treating forms as generic documents obscures the core difficulty.

### 3.2   TERMINOLOGY DEFINITION

We then begin by providing definitions for the terminology used in FORMCRAFT:

- **Level** ($\mathbb{L}$): A hierarchical component of FORMCRAFT taxonomy representing a distinct aspect of form understanding, organized into three levels—content modality, layout structure, and semantic relation—with increasing complexity and abstraction.
- **Field**: The elemental data collection component in a form, representing the smallest indivisible unit where users provide information through various input mechanisms such as checkboxes, radio buttons, and text entry areas.
- **Group**: A coherent collection of fields bound by functional purpose and typically spatial contiguity, serving a unified information-gathering objective.
- **Modality**: Perceptual mode of information representation within fields that determines both presentation and interaction patterns, comprising textual, marking, visual, and cross-modal categories with distinct recognition requirements.
- **Modality subtype**: The specific variant or implementation of a modality that defines the precise format of information representation, such as machine-printed or handwritten text within textual modality.

### 3.3   $\mathbb{L}1$: CONTENT MODALITY

We define content modality as the perceptual channel through which information is represented in form fields. This first level of taxonomy is fundamental as it directly impacts how vision-language models processes form content. We categorize form fields into four distinct modalities based on their representation characteristics, each presenting unique recognition challenges: **Textual, Marking, Visual, and Cross-modality**.

Textual modality encompasses machine-printed text, handwritten text, mixed formats, and text embedded in drawings. Marking modality includes interactive elements like checkboxes, tables/grids, and annotations/markings that modify content. Visual modality covers static images, charts/diagrams, hand-drawn sketches, and machine-printed paint elements like stamps. Cross-modality ad-

dresses scenarios where multiple modalities interact within a single field, requiring models to process each modality while understanding their spatial and semantic relations. Detailed descriptions of each modality subtype, including their unique characteristics and challenges, are provided in the supplementary materials.

### 3.4 L2: LAYOUT STRUCTURE

Layout Structure Level addresses the hierarchical organization of forms, from simple single-level layouts to complex nested blocks with inter-field relationships. In this framework, we define three primary layout structures: (a) Field: the basic unit of information containing individual form elements as defined in content Level. (b) Group: a collection of semantically and spatially related fields that form a logical unit, often sharing properties such as a common title or purpose. (c) Nested: complex structures where groups can contain other groups, creating a hierarchical organization that reflects the form's logical structure. This nested architecture allows for representing sophisticated form layouts while maintaining clear relationships between components.

### 3.5 L3: SEMANTIC RELATION

In structured data extraction, cross-reference understanding ensures accuracy and coherence across interdependent fields in real-world forms like medical records, legal documents, and financial statements. FORMCRAFT provides the following taxonomy to classify relationship among fields of a form:

- **Reference boundary** identifies where relationships occur within the structure of the form: (a) Intra-Group: Cross-references within the same group (e.g., text-to-text or checkbox-to-handwritten note within the same section). (b) Inter-Group: Cross-references spanning different groups or sections on the same page (e.g., text-to-text or visual diagram-to-table field across different blocks). (c) Cross-Page/Document: Relationships extending across multiple pages or external documents (e.g., ID text on page 1 referencing text on page 3).
- **Semantic linkage** defines how fields or elements are logically linked: (a) Direct Link: A field's value, state, or selection influences another field's state (e.g., a checkbox enabling/disabling a table field). (b) Consistency or Constraint: Fields must maintain logical consistency (e.g., a handwritten "No Allergy" note must align with a radio button labeled "No Allergy"). (c) Domain-Specific / Knowledge-Driven: Relationships requiring specialized knowledge (e.g., medical codes linking to diagram annotations of procedure sites).
- **Dependency complexity** measures the intricacy of referencing: (a) Simple: A single condition links one field to another (e.g., a checkbox toggles a visual signature box). (b) Compound: Multiple parallel relationships exist simultaneously (e.g., multiple numeric fields must match a total). (c) Chained: One reference depends on another in a multi-step chain (e.g., selecting an option in Section A reveals a table in Section B, which then necessitates a visual annotation in Section C).

## 4 BENCHMARK DESIGN AND EVALUATION

### 4.1 DATASET AND ANNOTATION

We evaluate on a fully de-identified, privacy-compliant dataset consisting of 101 open-source annotated samples (OmniAI, 2025) and 265 proprietary dental forms, selected for their diverse multi-modal complexity—textual, marking, visual, and mixed inputs. We annotated these samples with modality and input types, capturing logical structure and enabling comprehensive evaluation across field-level, layout, and semantic relations. To test generality, we additionally demonstrate how to apply FORMCRAFT to OCRBench v2 (Fu et al., 2024) as detailed in the supplementary.

### 4.2 MODALITY-LEVEL TASKS AND EVALUATION

Modality-level evaluation assesses the model's proficiency in accurately recognizing content within individual fields. This task encompasses: (a) OCR for both printed and handwritten text. (b) Detection of checkboxes and circles for marking-type fields. (c) Recognition of cross-modal content

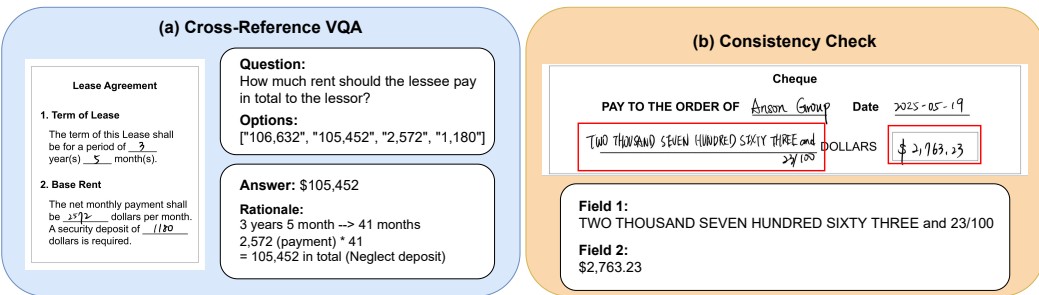

Figure 3: These two examples illustrate scenarios that require the model to perform cross-referencing. In (a), the model must integrate information from multiple fields to answer the question accurately. In (b), consistency check evaluates whether the model can correctly extract information from all relevant fields.

within individual fields, ensuring that multi-modal entries (e.g., text inside a checkbox-labeled section) are correctly interpreted.

The evaluation of recognition accuracy is performed using the following metrics:

- Marking-type fields are evaluated using the F1-score, which captures precision and recall.
- Textual fields are assessed using a combination of WER, CER, and BLEU scores to quantify recognition performance.
- Mixed-modality fields require all constituent components within a field to be accurately recognized for the field to be deemed correct, ensuring a stringent evaluation for multi-modal dependencies.

### 4.3 LAYOUT-LEVEL TASKS AND EVALUATION

Layout-level evaluation focuses on assessing the model's ability to reconstruct the hierarchical organization of form elements. This includes: (a) Grouping individual fields into coherent logical blocks. (b) Identifying nested structures and multi-layer hierarchies. (c) Accurately reconstructing document layouts while preserving spatial relationships.

The structural accuracy of the predicted layout is measured using Tree-Edit-Distance (TED) (Zhang & Shasha, 1989) with depth-sensitive cost functions, which compares predicted and ground-truth layout trees. The cost function assigns lower penalties for modifications at deeper hierarchical levels, thereby emphasizing high-level layout accuracy. We provide the following cost functions:

$$C_{\text{delete}}(n) = \frac{1}{1 + d_n} \tag{1}$$

$$C_{\text{insert}}(n) = \frac{1}{1 + d_n} \tag{2}$$

$$C_{\text{substitute}}(n_1, n_2) = \begin{cases} 0 & \text{if } l_{n_1} = l_{n_2} \\ \frac{1}{1 + (d_{n_1} + d_{n_2})/2} & \text{otherwise} \end{cases} \tag{3}$$

where $d_n$ represents the depth of node n and $l_n$ represents the node's label. This depth-weighted approach assigns lower costs to modifications at deeper levels of the tree, reflecting the intuition that structural changes near the root have more significant impact on overall layout understanding.

### 4.4 SEMANTIC RELATION TASKS AND EVALUATION

To examine the model's capability to capture logical dependencies across different fields, FORM-CRAFT framework provides two kinds of evaluation approaches: (a) QA-style metrics: A VQA framework generates structured queries to verify semantic relation. (b) Consistency checks: Ensures logically linked fields uphold valid dependencies based on the reference scope, reference type and complexity.

Table 1: **Textual modality (L1)** We assessed textual modality across four models using two input types: machine-printed and handwritten text. The taxonomy clearly demonstrates that handwritten OCR remains more challenging than printed text OCR.

| Methods | Machine-Printed | | | Handwritten | | |
|---|---|---|---|---|---|---|
| | CER ↓ | WER ↓ | BLEU ↑ | CER ↓ | WER ↓ | BLEU ↑ |
| GPT-5 (Achiam et al., 2023) | 0.21 | 0.29 | 0.15 | 0.41 | 0.55 | 0.03 |
| Claude 4 Sonnet (Anthropic, 2025) | **0.20** | **0.25** | **0.17** | 0.35 | **0.40** | **0.05** |
| Gemini 2.5 Flash (Comanici et al., 2025) | 0.22 | 0.31 | 0.15 | **0.34** | 0.42 | 0.04 |
| Qwen2.5-VL-72B (Bai et al., 2025) | 0.25 | 0.31 | 0.16 | 0.39 | 0.48 | 0.04 |

Table 2: **Marking modality (L1)** We assessed marking modality across four models using two input types: checkboxes and circles. Results indicate that models generally find circle markings more challenging to identify accurately than checkboxes.

| Methods | Checkbox | | | Circle | | |
|---|---|---|---|---|---|---|
| | F1 ↑ | Prec. ↑ | Rec. ↑ | F1 ↑ | Prec. ↑ | Rec. ↑ |
| GPT-5 (Achiam et al., 2023) | 0.59 | 0.58 | 0.62 | 0.61 | 0.73 | 0.65 |
| Claude 4 Sonnet (Anthropic, 2025) | **0.75** | **0.78** | **0.79** | **0.66** | **0.74** | 0.68 |
| Gemini 2.5 Flash (Comanici et al., 2025) | 0.62 | 0.51 | 0.71 | 0.63 | 0.55 | **0.81** |
| Qwen2.5-VL-72B (Bai et al., 2025) | 0.59 | 0.56 | 0.72 | 0.51 | 0.63 | 0.46 |

**Cross-reference VQA**   Present a framework in FORMCRAFT for generating VQA samples that test a model's ability to cross-reference multiple fields in a form. As illustrated in Fig. 3 (a), the example involves the question "How much rent should the lessee pay in total to the lessor?" To answer correctly, the model must first consult the "Term of Lease" group, where describing the total period of lease, then calculates the total price of lease. We observe that models usually include the security deposit in the calculation and resulting in a wrong prediction. We provide a failure example in the supplementary. This multi-step reasoning process mirrors real-world document interpretation where critical information is distributed across different sections rather than contained in a single field. The approach challenges models to develop stronger spatial reasoning and contextual understanding—capabilities essential for practical document processing systems. We provide ground-truth annotations to guide LLMs in generating test samples that systematically probe these cross-referential abilities across varying levels of complexity. Please refer to the supplementary for the VQAs curation process.

**Consistency checks**   Implement consistency checks in FORMCRAFT to assess models' ability to cross-reference related fields within forms. This framework identifies field pairs requiring aligned content, tasking models to recognize and reconcile information across them, as shown in Fig. 3 (b). This mimics human reasoning, inferring answers from context despite unclear annotations. Beyond recognition accuracy, it tests contextual reasoning, exposing gaps between models and humans, especially with complex visual relationships or spatially distant elements. These checks offer a nuanced evaluation of visual-linguistic reasoning, surpassing basic OCR metrics.

## 5 EXPERIMENT AND ANALYSIS

### 5.1 IMPLEMENTATION DETAILS

We use a three-level taxonomy—content modality, layout structure, and semantic relation—to evaluate form understanding across diverse field types. Fields in the 366-form dataset are classified into textual, marking, visual, and cross-modality types using the method from Sec. 3.2. For each modality, we apply the metrics in Sec. 4.2 to assess model performance, enabling detailed analysis tailored to each field's unique challenges.

(a) **Visual modality** ($\mathbb{L}1$) In FORMCRAFT framework, diagram subtype can be extracted into text information and evaluated. Results indicate that most LVLMs struggle with low performance.

| Methods | CER ↓ | WER ↓ | BLEU ↑ |
|---|---|---|---|
| GPT-5 | 0.19 | 0.31 | 0.20 |
| Claude 4 Sonnet | **0.18** | **0.21** | **0.23** |
| Gemini 2.5 Flash | 0.24 | 0.31 | 0.18 |
| Qwen2.5-VL-72B | 0.28 | 0.30 | 0.11 |

(b) **Cross-modality** ($\mathbb{L}1$) In FORMCRAFT, we test models to accurately extract all content in a field that contains multiple modalities, such as checkboxes with accompanying text.

| Methods | Accuracy ↑ |
|---|---|
| GPT-5 | 0.81 |
| Claude 4 Sonnet | **0.86** |
| Gemini 2.5 Flash | 0.77 |
| Qwen2.5-VL-72B | 0.78 |

Table 3: Results for visual and cross-modality evaluations.

## 5.2 BASELINE MODELS

To evaluate the effectiveness of our FORMCRAFT benchmark, we test several state-of-the-art VLMs as baselines, assessing their performance across the three-level taxonomy: content modality, layout structure, and semantic relation. The selected models include GPT-5 (Achiam et al., 2023), Claude 4 Sonnet (Anthropic, 2025), Gemini 2.5 Flash (Comanici et al., 2025), Qwen 2.5-VL-72B-Instruct (Bai et al., 2025).

## 5.3 EVALUATION RESULTS

$\mathbb{L}1$ **evaluation** We evaluate baseline models across subtypes of the FORMCRAFT framework's $\mathbb{L}1$ modality.

In textual tasks (Table 1), Claude 4 Sonnet leads in printed text recognition, while Qwen 2.5-VL-72B-Instruct shows stronger performance on handwriting. GPT-5 handles printed text well but struggles with handwriting, and Gemini 2.5 Flash improves on handwriting but performs poorly in reconstruction.

In marking tasks (Table 2), Claude 4 Sonnet excels in checkbox and circle detection, with Qwen close behind in precision. Gemini 2.5 Flash emphasizes recall, while GPT-5 remains inconsistent.

For visual modality (Table 3a), Claude 4 Sonnet achieves the best overall accuracy, GPT-5 shows strong character-level precision, Qwen remains balanced, and Gemini 2.5 Flash lags behind.

Cross-modality evaluation (Table 3b) confirms Claude 4 Sonnet's robustness in integrating diverse inputs, followed by GPT-5 and Qwen, with Gemini trailing.

Overall, Claude 4 Sonnet is the strongest across modalities, Qwen 2.5-VL-72B-Instruct is resilient on handwriting, while GPT-5 and Gemini 2.5 Flash show more weaknesses. These results highlight the challenge of generalizing across modalities and demonstrate the value of FORMCRAFT 's granular evaluation.

$\mathbb{L}2$ **evaluation** In Table 4a, we present the evaluation results for layout structure ($\mathbb{L}2$) level of the FORMCRAFT framework, as detailed in Sec. 4.3.

Claude 4 Sonnet outperforms the other models, demonstrating the closest alignment with the true layout structure. This suggests a superior capability to capture spatial relationships and hierarchical arrangements in complex forms. GPT-5 follows as a reasonable contender, though it exhibits a noticeable drop in precision compared to Claude, indicating some inconsistencies in layout interpretation. In contrast, Gemini 2.5 Flash and Qwen 2.5-VL-72B-Instruct struggle significantly, with performances that diverge further from the ground truth. This gap highlights their limitations in processing intricate structural details, potentially due to weaker spatial reasoning or sensitivity to layout variability.

These results underscore the importance of robust layout comprehension in form processing tasks, where precise element positioning is critical for downstream applications. Claude 4 Sonnet's strong performance reinforces its overall reliability across the FORMCRAFT framework, while the challenges faced by Gemini and Qwen suggest that layout structure remains a differentiating factor

(a) **Layout structure** (𝕃2) The experiments demonstrate that even the most advanced proprietary LVLMs still struggle to reconstruct the hierarchical structure of complex forms. More generated examples are provided in the supplementary materials.

(b) **Semantic relation** (𝕃3) The experiments show that while current models excel in VQA, even for advanced reasoning questions, they struggle with consistency in cross-referencing scenarios. This suggests potential biases toward certain task formats while lacking proficiency in others.

| Methods | Distance↓ |
|---|---|
| GPT-5 | 45.15 |
| Claude 4 Sonnet | **30.20** |
| Gemini 2.5 Flash | 60.15 |
| Qwen2.5-VL-72B | 76.06 |

| Methods | Cross-Ref. VQA(%)↑ | Consistency Check(%)↑ |
|---|---|---|
| GPT-5 | **79.13** | 36.52 |
| Claude 4 Sonnet | 77.69 | **50.13** |
| Gemini 2.5 Flash | 69.55 | 43.18 |
| Qwen2.5-VL-72B | 60.06 | 45.00 |

Table 4: Results for layout structure and semantic relation evaluations.

among state-of-the-art models. The findings validate the inclusion of the 𝕃2 level in our evaluation methodology, emphasizing its role in assessing a model's holistic understanding of document organization.

**𝕃3 evaluation** In Table 4b, we present the evaluation results for semantic relation (𝕃3) level of the FORMCRAFT. GPT-5 demonstrates a strong capability in cross-referential understanding, excelling at linking related elements across the form, though it falters significantly in maintaining consistency, suggesting a disconnect between comprehension and coherence. Claude 4 Sonnet, while slightly less adept at cross-referencing compared to GPT-5, achieves a more balanced performance by outperforming others in consistency checks, indicating a better grasp of semantic integrity. Gemini 2.5 Flash and Qwen 2.5-VL-72B-Instruct lag behind in cross-referential accuracy, with Qwen showing a modest edge in consistency over Gemini, though neither approaches Claude's level.

These results reveal substantial gaps in current VLMs for complex form understanding, particularly in capturing and maintaining semantic relation. The disparity between cross-referential accuracy and consistency underscores a critical challenge: while some models can identify connections, ensuring logical coherence across responses remains elusive. This affirms the value of FORMCRAFT in exposing such deficiencies, highlighting the need for improved reasoning and relational understanding in form processing tasks. Claude 4 Sonnet emerges as the most reliable in this level, yet even its performance indicates room for advancement in handling the nuanced semantics of structured documents. More analyses are provided in the supplementary.

## 6 CONCLUSION AND FUTURE WORK

We present FORMCRAFT, a benchmarking framework for evaluating the form understanding capabilities of large vision-language models. FORMCRAFT three-level taxonomy enables rigorous assessment based on (a) Content modality, (b) Hierarchical layout structure, and (c) Semantic relation—progressing from fundamental content recognition and layout reconstruction to sophisticated cross-referencing reasoning. This systematic approach provides fine-grained insights into model performance across diverse document types and complexity levels, revealing specific strengths and limitations in form understanding tasks.

FORMCRAFT evaluation addresses current gaps in form intelligence assessment by isolating specific capabilities and providing targeted performance metrics. Experiments demonstrate that even state-of-the-art models encounter significant challenges with complex layouts, mixed modalities, and inferential reasoning across form elements. By releasing FORMCRAFT, we aim to catalyze a necessary shift in document AI research: moving from text-centric extraction tasks to the more challenging and practical frontier of structure- and relation-centric understanding. Our work provides the tools to measure progress on this new frontier. For future work, we plan to curate an even more diverse collection of accessible yet complex real-world form data to establish a comprehensive benchmark, thereby providing a new evaluation standard for the research community.

# 7 ETHICS STATEMENT

All data used in this work were fully de-identified prior to annotation and analysis to protect the privacy of individuals. No personally identifiable information or sensitive attributes were retained in the dataset. Annotation was conducted with professional standards to ensure data quality and integrity. Our work adheres to the ICLR Code of Ethics, particularly with respect to privacy preservation and the responsible release of research artifacts.

# 8 REPRODUCIBILITY STATEMENT

As this work introduces a benchmark rather than a novel model, we have prioritized transparency and reproducibility. Upon acceptance, we will release the majority of the dataset, the annotation schema, and accompanying evaluation scripts to facilitate fair comparison across models. A private test set will be retained to prevent overfitting and to preserve the validity of leaderboard evaluations. Further details regarding dataset construction and task definitions are described in the main text and supplementary materials.

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

## A LIMITATION

Our taxonomy effectively captures semantic relationships within documents but has limitations in evaluating spatial understanding capabilities. While FORMCRAFT provides a framework for assessing content modality, hierarchical structure, and semantic relations, it does not fully measure how models comprehend the spatial arrangement of elements in documents. This limitation means our evaluation framework may not completely assess a model's processing of document layout and positioning, which represents an additional dimension of form understanding beyond the semantic connections our taxonomy currently addresses.

## B LLM USAGE STATEMENT

We employed LLMs in three contexts: (1) for inference experiments to evaluate existing models on our benchmark; (2) to generate semantic VQA questions as part of the dataset construction; and (3) to assist in refining the grammar and semantics of the manuscript. The scientific contributions and claims of this work are solely those of the authors, and we take full responsibility for all content.

## C POTENTIAL NEGATIVE SOCIETAL IMPACTS

FORMCRAFT framework's reliance on datasets could perpetuate biases if the data lack diversity, leading to unequal performance across different cultural and linguistic groups. Although FORM-CRAFT itself is not directly responsible for job displacement, the broader adoption of form automation technologies it supports may contribute to changes in employment dynamics. Nonetheless, with appropriate ethical guidelines and thoughtful implementation, these challenges can be effectively managed to maximize the benefits of advanced form processing technologies.

## D DETAIL DEFINITIONS OF ALL MODALITY SUBTYPES

**Textual modality**  Textual information plays a crucial role in form understanding, with various representations affecting model performance. We categorize this modality into four common modality subtypes: Machine-Printed Text, which is standardized and structured (field labels, instructions, pre-filled content), typically well-defined but varying in font style, size, or orientation; Handwritten Text, which introduces variability through different styles, cursive forms, and potential ambiguities, requiring effective recognition for processing manually completed forms; Mixed Printed and Handwritten Text, which combines machine-printed labels with handwritten responses, creating challenges in segmentation, classification, and recognition as models must distinguish between these different elements; and Text in Drawings, where information embedded within diagrams, sketches, or annotations appears in various orientations and styles, sometimes integrated with complex graphical elements, making accurate detection and extraction difficult.

**Marking modality**  Interactive and marking elements contribute significantly to the structural and semantic complexity of documents in form understanding. We categorize this modality into three subtypes: Checkboxes and Radio Buttons, which are selection-based inputs indicating user choices in multiple-choice questions, medical forms, and surveys, requiring detection, label association, and state recognition; Tables and Grids, which present information in a row-column format across invoices, registration forms, and financial statements, demanding accurate parsing of table structures, header-value relationships, and handling of alignment and formatting variations; and Annotations and Markings, which include handwritten or graphical elements like underlining, circling, or sticky notes that modify existing content, introducing ambiguities requiring contextual reasoning beyond traditional OCR approaches.

**Visual modality**  Visual elements play a crucial role in form understanding, requiring models to process and interpret various graphical information types. We categorize visual modality into four subtypes: Static Images and Illustrations, including photographs, product illustrations, and machine-readable codes, which require both detection and content interpretation; Charts and Diagrams, such as flowcharts and statistical graphs that convey structured information graphically, demanding recognition of both visual structure and semantic relations; Hand-Drawn Sketches, including user-created

drawings, arrows, or symbols that introduce complexity through their informal nature and representational variance; and Machine-Printed Paint, comprising mechanically applied elements like postmarks and stamps that serve as official validation or temporal indicators on forms.

**Cross modality** Forms often contain complex scenarios where multiple modalities coexist and interact within a single field. Cross-modality encompasses any combination of the textual, marking, and visual modalities described above. For example, forms may contain images with handwritten annotations (combining visual and textual modalities) or tables with both checkboxes and icons (combining marking and visual modalities). These cross-modal scenarios present unique challenges as they require models to not only process each modality independently but also understand the semantic and spatial relationships between different modal elements.

# E FORMCRAFT ON OCRBENCH V2

To demonstrate FORMCRAFT framework's generality, we evaluate it within OCRBench v2 (Fu et al., 2024), a benchmark assessing optical character recognition and multimodal reasoning across 29 datasets. We focus on FUNSD (Jaume et al., 2019) (document) and SROIE (Huang et al., 2019) (receipt), both featuring human annotations, to demonstrate FORMCRAFT's ability to address diverse form understanding challenges through its leveled, modality-specific approach. In Fig. 4, we present an example image from FUNSD—a receipt with handwritten text and annotations. The receipt originally contained two items priced at 4.20\$ and 0.95\$, with a subtotal of 5.15\$. However, one item was circled and another was marked as canceled, resulting in a final price of 4.20\$, shown as a handwritten modification over the original 5.20\$ in the "CASH" field. This example demonstrates complex semantic relations that FORMCRAFT can clearly define through its taxonomy of modalities, subtypes, hierarchical structure, and semantic relations. With the FORMCRAFT framework, it's possible to design more advanced evaluations requiring models to demonstrate hierarchical structure understanding and cross-referencing reasoning. For example, the original question provided in (Huang et al., 2019) is "Find out the company name, date, address, and total amount issued in this receipt." and our generated cross-referencing question might be "Q: Which item corresponds to the total cash payment? A: ITEM # 9555501403092".

**FUNSD** Comprising with 50 noisy scanned forms in OCRBench v2 (Fu et al., 2024), it is annotated for text, layout, and entity relationships, making it a robust testbed for form comprehension. Its standard evaluation emphasizes text detection, entity classification (e.g., headers, questions, answers), and pairwise linking, yet it often overlooks the hierarchical structure and semantic coherence that FORMCRAFT targets. In the $\mathbb{L}1$, we would assess recognition accuracy across printed and handwritten fields, leveraging FUNSD's (Jaume et al., 2019) noise to probe model robustness—an aspect underexplored in its typical metrics. For the $\mathbb{L}2$, FORMCRAFT would analyze the spatial hierarchy of form elements, such as the arrangement by using the concepts of field, group and nested-group and calculating the TED, which FUNSD (Jaume et al., 2019) annotations provide but do not fully evaluate for structural fidelity. In the $\mathbb{L}3$, we would test cross-referential understanding and consistency, areas where FUNSD's (Jaume et al., 2019) linking annotations fall short of assessing contextual dependencies or logical coherence across the document. This approach highlights deficiencies in current VLMs, such as those observed in our prior $\mathbb{L}3$ evaluations, where models excel in isolation but falter in relational tasks.

**SROIE** With 125 scanned receipt image in OCRBench v2 (Fu et al., 2024), focuses on OCR and key information extraction, annotating four fields: company, date, address, and total. While SROIE's (Huang et al., 2019) evaluation excels at text localization and field detection, it prioritizes surface-level accuracy over structural and semantic depth, aligning well with FORMCRAFT 's mission to uncover such gaps. In the $\mathbb{L}1$, we would evaluate text recognition, testing resilience beyond basic OCR—a challenge its metrics underexplore. For the $\mathbb{L}2$, FORMCRAFT would examine the spatial organization of receipt elements, such as the positioning of item lists relative to totals, which SROIE (Huang et al., 2019) annotations capture but do not assess for structural integrity. In the $\mathbb{L}3$, FORMCRAFT would scrutinize the consistency and contextual linkage of extracted fields, an area SROIE's (Huang et al., 2019) key extraction task does not fully address. For instance, we identified a receipt in the SROIE (Huang et al., 2019) dataset with visible correction marks, where the total field reflects an outdated amount superseded by modifications elsewhere on the document.

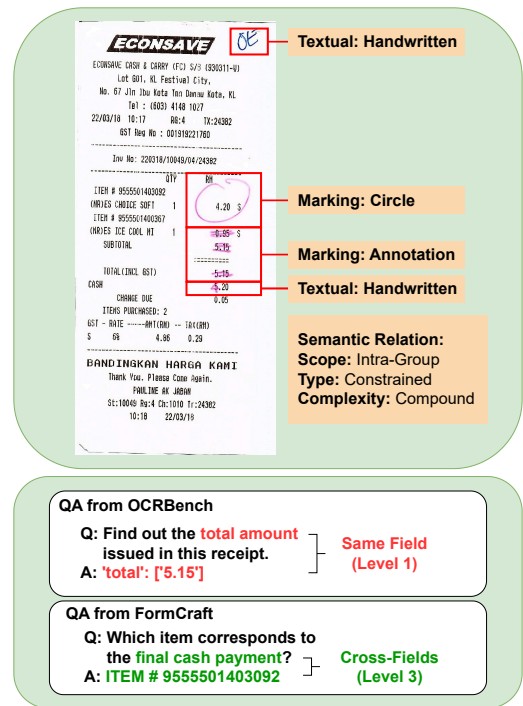

Figure 4: This receipt image from (Fu et al., 2024) demonstrates how the framework can be adapted to any existing form-like dataset. Compared to existing datasets that only focus on extracting partial information, the evaluation framework requires more advanced reasoning abilities across different fields or groups within a form.

In a VQA task, a model should output the correct final total by interpreting the entire form's content modality—including annotation traces from the marking modality (e.g., correction marks) and machine-printed text from the textual modality—alongside the receipt's overall structure. However, OCRBench v2's (Fu et al., 2024) SROIE (Huang et al., 2019) evaluation typically limits VQA to straightforward text queries, such as directly reading the total field, which yields an incorrect amount in this case. This approach fails to test a model's true ability to integrate multimodal cues and reason about semantic coherence. Our prior evaluations suggest that VLMs like GPT-4o (Achiam et al., 2023) and Claude 3.5 Sonnet (Anthropic, 2025) exhibit uneven performance in such relational tasks, and FORMCRAFT 's leveled analysis would expose these limitations, highlighting the need for deeper contextual understanding beyond SROIE's (Huang et al., 2019) standard metrics.

## F   VQA GENERATION

We provide the prompts and codes we used to generate VQA samples and consistency check samples.

### F.1   STEP 1: DATA CONVERT

The first step is to convert a annotated json file into the structure format (fields, group, ...) as our definition in FORMCRAFT. Also, we use the GPT-4.1 to extract the modality and subtype-modality information. The prompt is provided here:

```
"""
You are provided with an image and the annotated content of the image.
Your task is to convert the provided information to the JSON file according to the
    following taxonomy:
```

```
Term definitions:
Field: The elemental data collection component in a form, representing the smallest
    indivisible unit where users provide information through various input mechanisms
    such as checkboxes, radio buttons, and text entry areas.
Group: A coherent collection of fields bound by functional purpose and typically
    spatial contiguity, serving a unified information-gathering objective.
Modality: Perceptual mode of information representation within fields that determines
    both presentation and interaction patterns, comprising textual, marking, visual,
    and cross-modal categories with distinct recognition requirements.
Modality subtype: The specific variant or implementation of a modality that defines the
    precise format of information representation, such as machine-printed or
    handwritten text within textual modality.

Taxonomy:
Content Modality
 - Textual: Machine-Printed, Handwritten, TextInDraw, Mixed
 - Marking: Checkbox, Radio Buttons, Table/Grid, Annotations
 - Visual: Statistic Image, Chart, Diagram, Hand-Drawn, Printed Paint
 - Cross: Any combination of the above modalities

You should identify the fields and groups in the image and convert them to the JSON
    format.
"""
```

Listing 1: Format convertion prompt.

Additionally, we control the output format by providing the output structure definition (FormCraft class) to GPT-4.1:

**Defined class code:**

```
class FieldModel(BaseModel):
    label: str = Field(..., description="The label of the field.")
    modality: str = Field(..., description="The modality of the field.")
    modality_subtype: str = Field(..., description="The modality subtype of the field.")
    value: str = Field(..., description="The value of the field.")

class Group(BaseModel):
    label: str = Field(..., description="The label of the group.")
    fields: List[FieldModel] = Field(..., description="The fields in the group.")
    # This allows groups to contain nested groups.
    groups: Optional[List["Group"]] = Field(None, description="The nested groups within
     the group.")

class FormCraft(BaseModel):
    fields: List[FieldModel] = Field(..., description="The fields in the form.")
    groups: List[Group] = Field(..., description="The groups in the form.")
```

## F.2 STEP 2: HUMAN VERIFICATION

We perform human verification on all the 366 converted data obtained from Step 1 to make sure that the generated results meet our requirement and align with the content presented in the images.

## F.3 STEP 3: VQA GENERATION

We provide the prompt we use to generation VQA from the annotated information.

```
"""
You are provided with a taxonomy that defines cross-reference relationships within form
    annotations. The taxonomy is structured along three levels:

1. Reference Scope:
   - Intra-Group: Cross-references within the same group (e.g., a checkbox in one field
     affecting a text field in the same block).
```

```
        - Inter-Group: Cross-references between different groups/sections on the same page
          (e.g., a text field in one section linking to a diagram in another).
        - Cross-Page/Document: Relationships spanning multiple pages or external documents
          (e.g., an ID on page 1 linking to text on page 3).

    2. Reference Type:
        - Direct Link: A single fields value or state directly enables or modifies another
          field.
        - Consistency or Constraint: Fields must be logically consistent or satisfy
          constraints (e.g., numeric fields summing to a total).
        - Domain-Specific / Knowledge-Driven: References that require specialized rules or
          domain knowledge (e.g., billing or medical codes referencing related annotations).

    3. Reference Complexity:
        - Simple: A single condition linking two fields.
        - Compound: Multiple parallel relationships where several fields must meet shared
          constraints.
        - Hierarchical (Chained): A multi-step chain of dependencies (e.g., Field A triggers
          Field B, which in turn triggers Field C).

    Additionally, each cross-reference may involve different modalities (e.g., text,
        checkbox, handwritten note, diagram):
    - Modal Relationship:
      - Single-Modality: Both elements are of the same type (e.g., text  text).
      - Cross-Modality: Different types are involved (e.g., a checkbox triggering a textual
        input).

    You are provided with a JSON object containing the form information annotated by
        professionals.
    You are tasked with generating some QA samples which might have cross-reference
        relationships according to the provided taxonomy.
    Specifically, to answer the question, the model needs to find the fields which might
        have special relation according to the provided taxonomy.
    """
```

Listing 2: VQA generation prompt.

We first collect some potential questions and select those with high quality and require cross-reference. We further provide these selected VQA samples as in-context examples and re-generate VQA samples. All after these procedures, we conduct human verification to preserve the quality of generated data.

### F.4  STEP 4: CONSISTENCY CHECK SAMPLE GENERATION

Meanwhile, we provide the prompt we use to generate samples we use in consistency check evaluation.

```
'''
You are provided with a taxonomy that defines cross-reference relationships within form
    annotations. The taxonomy is structured along three levels:

1. Reference Scope:
    - Intra-Group: Cross-references within the same group (e.g., a checkbox in one field
      affecting a text field in the same block).
    - Inter-Group: Cross-references between different groups/sections on the same page
      (e.g., a text field in one section linking to a diagram in another).
    - Cross-Page/Document: Relationships spanning multiple pages or external documents
      (e.g., an ID on page 1 linking to text on page 3).

2. Reference Type:
    - Direct Link: A single fields value or state directly enables or modifies another
      field.
    - Consistency or Constraint: Fields must be logically consistent or satisfy
      constraints (e.g., numeric fields summing to a total).
```

```
  - Domain-Specific / Knowledge-Driven: References that require specialized rules or
    domain knowledge (e.g., billing or medical codes referencing related annotations).

3. Reference Complexity:
  - Simple: A single condition linking two fields.
  - Compound: Multiple parallel relationships where several fields must meet shared
    constraints.
  - Hierarchical (Chained): A multi-step chain of dependencies (e.g., Field A triggers
    Field B, which in turn triggers Field C).

Additionally, each cross-reference may involve different modalities (e.g., text,
    checkbox, handwritten note, diagram):
- Modal Relationship:
  - Single-Modality: Both elements are of the same type (e.g., text to text).
  - Cross-Modality: Different types are involved (e.g., a checkbox triggering a textual
    input).

You are provided with a JSON object containing the form information annotated by
    professionals.
You are tasked with finding fields which might have special relation according to the
    provided taxonomy.
List some special relations and provide a detailed explanation of the reasoning behind
    each relation.
Provide a confidence score from 0.0 to 1.0 for each relation, indicating how certain
    you are about the relationship.

Identify pairs of fields with a special relationship where the value of one field can
    be directly determined or inferred based on the other fields value, even if that
    field is not filled in.

Exclude cases where:
 - The fields are only logically constrained (e.g., must match or be consistent).
 - The fields are related statistically or contextually, but the value cannot be
    computed or inferred directly.

Focus on:
 - Direct enabling relationships (e.g., a checked box triggers a specific narrative)
 - Value propagation (e.g., a calculated field, or a conditionally filled field)

If there is no such relationship, return an empty list.
'''
```

Listing 3: Consistency check samples.

# G   CASE STUDY 1

We provide a real example comparing the four baseline models. In this example, models were required to extract handwritten textual content from the form in Fig. 5-(b), with results as follows:

- Claude 3.5 Sonnet (Anthropic, 2025): "Process ↑ valplast with valplast clasp and V acrylic."

- GPT-4o (Achiam et al., 2023): "Process T Valplast with Valplast clasps and V acrylic."

- Gemini 2.0 Flash (Team et al., 2023): "Process ↑ Valplast with velplinot clasps and & acrylic."

- Qwen-2.5 VL 72B(Bai et al., 2025): "Process ↑ Uplift with Valplast clasps and ↓ acrylic."

- ground-truth: "Process ↑ Valplast with Valplast clasps and ↓ acrylic."

In this case, the four content items "UPPER", "LOWER", "VALPLAST CLASP" and "'ACRYLIC" appear elsewhere in the form. However, models sometimes still fail to recognize the handwritten content despite having cross-referencing opportunities. For humans, even when handwritten words are unclear, it remains relatively easy to reference other sections and deduce the correct answer.

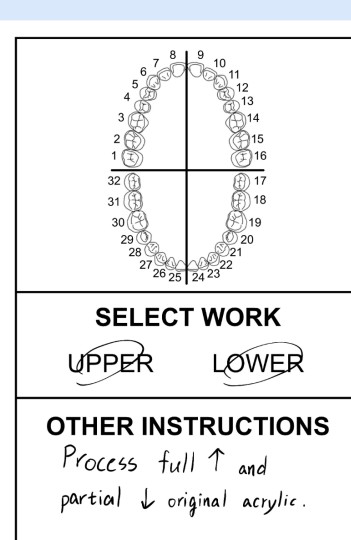

## (a) Cross-Reference VQA

**Question:** Which tooth might not involved in the procedure? Pick all that apply.

**Options:** ["Tooth 3", "Tooth 15", "Tooth 24", "Tooth 29"]

**Answer:** [ "Tooth 24", "Tooth 29"]

**Rationale:** "Process full upper and partial lower original acrylic", indicating that Tooth 24 and Tooth 29 might not be involved in the procedure.

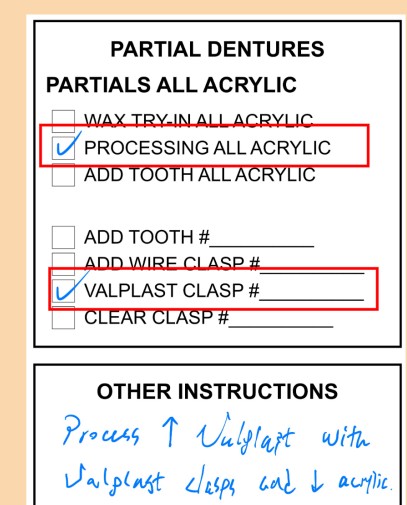

## (b) Consistency Check

**Wrongly Extracted (GPT-4o):** "Process ↑ Unlighit with Unlighit days add ↓ aorpiic.".

**Expected:** "Process ↑ Valplast with Valplast clasps add ↓ acrylic.".

**Rationale:** The possible correct word can be reference to the selected checkbox.

Figure 5: These two examples illustrate scenarios that require the model to perform cross-referencing. In (a), the model can refer to some other fields to answer the question. In (b), consistency check evaluates whether the model can correctly extract information from all relevant fields.

This observation suggests a potential direction for future research in advancing LVLM reasoning capabilities.

## H CASE STUDY 2

We provide another sample in which the GPT-4o failed. The question and provided form are presented in Fig. 6 and the response is provided in Fig. 7. We can find that inter-fields reasoning poses a challenge for current frontier MLLM.

## I   CASE STUDY 3

The original question from the SROIE (Huang et al., 2019) datasets:

```
{
    "image_path": "EN_part/kie_sroie/X51005605295.jpg",
    "question": "Find out the company name, date, address, and total amount issued in
    this receipt. Extract the required information using the text in the image
    directly, and return the result in a dict with keys 'company', 'date', 'address',
    'total'. Your answer should be in the JSON format:\n{\n    \"key1\": \"..\",  #
    The firt key and value\n    \"key2\": \"..\"     # The second key and value\n
    ..\n}\n",
    "dataset_name": "SROIE",
    "type": "key information extraction en",
    "id": 1249,
    "answers": [
        "{'company': ['ECONSAVE CASH & CARRY (FC) S/B'], 'date': ['22/03/18'],
    'address': ['LOT GO1, KL FESTIVAL CITY, NO. 67 JIN IBU KOTA TMN DANAU KOTA, KL'],
    'total': ['5.15']}"
    ]
}
```

Listing 4: Failed JSON Format

## J   LLAMA'S INSTRUCTION FOLLOWING

In our evaluation, we included LLaMA-3.2-90B-Vision-Instruct (Dubey et al., 2024) as one of our baseline models. However, we observed that LLaMA-3.2-90B-Vision (Dubey et al., 2024) consistently failed to adhere to the JSON file format specified for inference, unlike other VLMs, which successfully complied in most cases. Below, we present an example where LLaMA failed to generate a response in the correct format.

```
{
  "schemaVersion": "1.0",
      "invoice":{
          "fields":{
              label: "Name",
              value: "Unknown",
}
```

Listing 5: Failed JSON Format

This output is invalid JSON because it lacks a closing brace for the "`fields`" object and uses an unquoted key (`label` and `value`) without proper key-value pair syntax (should use "`label`": "Name").

Figure 6: We provide this example form to GPT-4o.

To calculate the total rent the lessee should pay to the lessor, we add the **total rent over the lease term** and the **security deposit**.

**Given:**

- **Lease Term**: 3 years and 5 months = $3 \times 12 + 5 = 41$ months

- **Monthly Rent**: $2,572

- **Security Deposit**: $1,180

**Total Rent:**

$$\text{Total Rent} = (41 \text{ months} \times \$2,572) + \$1,180 = 105,452 + 1,180 = \boxed{\$106,632}$$

**Final Answer:**

**$106,632** in total.

Figure 7: The GPT-4o wrongly answers the question. The correct answer should exclude the security deposit.

