# OpenReview forum: "FormCraft: Beyond Documents - Benchmarking Form Intelligence"
_ICLR.cc/2026/Conference — ICLR 2026 Conference Withdrawn Submission_

### Official Review · Reviewer_Ucby · 2025-10-28

**Soundness:** 1
**Presentation:** 2
**Contribution:** 1
**Rating:** 2
**Confidence:** 4

**Summary:**

The paper introduces a new framework for the evaluation of form understanding. The proposed framework works at three different levels: content, layout and semantic relations. The authors propose different metrics to evaluate form understanding at each of these three levels and describe a new dataset with annotations for the three levels. The paper shows how this evaluation framework can be used for the evaluation of SoA VLMs for the task of form understanding

**Strengths:**

- The paper provides and evaluation of SoA VLMs on the task of form understanding, proposing a new fine-grained annotated dataset for this task.

**Weaknesses:**

- The main claims of the paper are not well supported by the proposed methodology. The authors claim that they aim at overcoming the limitations of evaluations based on ocr and vqa, but some of their metrics are based on ocr (for content) and vqa (for semantic relations) metrics, contradicting their initial claim.
- I think that the proposed three-level evaluation framework is not much useful. The final goal of form understanding is being able to extract all relevant information including relations among the different fields. For that, a VQA-based evaluation as proposed in section 4.4 should be enough. Answering those types of questions already imply being able to recognize content and interpret layout structure, which will be implicitly evaluated. Diversifying the evaluation with so many metrics is a distraction from the ultimate goal of evaluating form understanding.
- The work does not contain any technical contribution that advances form understanding. Only SoA VLMs are evaluated without proposing any specific new baseline.
- There are some aspects of the definitions and terminology that, in my opinion, are not relevant or well defined. For instance, I do not understand why modality subtype is relevant, we should be able to recognize any content independently of the format (machine-printed, handwritten). Also, I do not understand how in a document form a checkbox can enable/disable a table field (definition of semantic linkage), ...

**Questions:**

- Concerning the metrics, why BLEU is used as an additional metric for evaluating content recognition? Why is not enough with CER and WER? The same for visual modality? What is evaluating visual modality? Why are metrics such as CER, WER and BLEU used to evaluate recognition of visual content?
- It is not clear what is the expected output of the evaluated models for the different tasks. For instance, for recognition is it just a sequence of the text in the form? How is it compared this output with the annotation? For the layout, which is the expected output? A layout tree? In which format? Maybe it would be helpful to provide some example of the prompts used as input to the VLMs to better illustrate how the VLMs operate and what is the expected output in each task.

---

### Official Review · Reviewer_NzMa · 2025-10-30

**Soundness:** 2
**Presentation:** 3
**Contribution:** 2
**Rating:** 4
**Confidence:** 4

**Summary:**

This paper introduces FormCraft, a comprehensive benchmark designed to evaluate the form intelligence of VLMs by moving beyond the limitations of simple OCR and VQA tasks. The authors propose a novel three-level taxonomy that assesses a model's performance on: content modality, layout structure and semantic relation. Through experiments on the annotated dataset, the study reveals that while current SOTA models are competent at basic content recognition, but failed in understanding complex structures and logical relationships. FormCraft thus highlights a critical gap in document AI and provides a standardized framework to guide future research toward more robust and structure-aware form processing.

**Strengths:**

The paper introduces a structured framework that redefines the evaluation of form intelligence, moving beyond simple text extraction to a more holistic, multi-layered assessment. Through rigorous experiments on this real-world dataset, the work provides a significant finding: even the most advanced VLMs excel at basic recognition, but fail substantially at understanding hierarchical structure and logical dependencies, highlighting a critical research gap. By identifying a key bottleneck and providing a comprehensive benchmark with a public dataset, the paper shifts the focus of the entire research community from solved OCR problems to the more challenging and practical form understanding.

**Weaknesses:**

The dataset heavily skewed towards a single domain, which significantly limits the generalizability of the benchmark's findings and its claim as a new standard.

**Questions:**

Does using GPT-4.1 to generate L3 test questions introduce a risk of model bias for the GPT family? Does the style of generated questions favor models from the same family?

---

### Official Review · Reviewer_6BY8 · 2025-10-31

**Soundness:** 1
**Presentation:** 2
**Contribution:** 1
**Rating:** 2
**Confidence:** 5

**Summary:**

This paper proposes a three-level taxonomy to tree the form understanding tasks from content modality, layout structure, and semantic relation understanding. A benchmark dataset is proposed to evaluate VLMs on three aspects. Overall, the contribution is limited from my perspective and may consider submitting to a dataset track after improving the contributions, paper structure, and analysis depth.

**Strengths:**

1. The author pointed out the gaps in AI-based form understanding.
2. A benchmark dataset is introduced, which might be helpful for future work.

**Weaknesses:**

The current version of the paper requires substantial revision and enhancement in the following aspects:

- Novelty and Contribution:
While the paper aims to conceptualize the form understanding task, it primarily introduces a benchmark without providing in-depth analysis or methodological advancements. The contributions need to be significantly extended to demonstrate meaningful progress beyond existing work.

- Dataset Quality and Analysis:
As a dataset paper, the current version lacks sufficient detail on annotation methodology, statistical distribution, and in-depth dataset analysis (e.g., per-type counts, fine-grained field categorization, and cross-page scenario handling). These aspects should be explicitly presented and discussed.

- Evaluation Scope:
The evaluation is limited to a few LLMs and does not explore performance across diverse model types, particularly Multimodal LLMs (MLLMs) in visually rich document domains. Moreover, incorporating results from established document-pretrained backbones would enhance the comprehensiveness of the benchmark. Some RAG and Agent-based backbone should also be considered.

- Task Definition and Clarity:
The definitions of the proposed task levels are not clearly articulated. A well-structured task definition diagram and corresponding explanations would help readers better understand the design and scope of the benchmark.

**Questions:**

Can the authors provide detailed dataset statistics and annotation procedures, including fine-grained field categories and cross-page scenarios?

Have the authors evaluated the benchmark with a wider range of models, including multimodal LLMs (MLLMs) and document-pretrained backbones?

---

### Note · Authors · 2025-12-02

I have read and agree with the venue's withdrawal policy on behalf of myself and my co-authors.